# Novel Starter Strain *Enterococcus faecium* DMEA09 from Traditional Korean Fermented *Meju*

**DOI:** 10.3390/foods12163008

**Published:** 2023-08-09

**Authors:** Seung-Eun Oh, Sojeong Heo, Gawon Lee, Hee-Jung Park, Do-Won Jeong

**Affiliations:** 1Department of Food and Nutrition, Dongduk Women’s University, Seoul 02748, Republic of Korea; 2Department of Food and Nutrition, Sangmyung University, Seoul 03016, Republic of Korea

**Keywords:** *Enterococcus faecium* strain DMEA09, *meju*, starter, protease, antibiotic susceptibility

## Abstract

The *Enterococcus faecium* strain DMEA09 was previously isolated from traditional Korean fermented *meju*. The objective of the current study was to investigate the traits of *E. faecium* strain DMEA09 as a starter candidate, focusing on its safety and technological properties. Regarding its safety, the DMEA09 strain was found to be sensitive to nine antibiotics (ampicillin, chloramphenicol, erythromycin, gentamicin, kanamycin, streptomycin, tetracycline, tylosin, and vancomycin) by showing lower minimum inhibitory concentrations (MICs) than the cut-off values suggested by the European Union Food Safety Authority for these nine antibiotics. However, its MIC value for clindamycin was twice as high as the cut-off value. A genomic analysis revealed that strain DMEA09 did not encode the acquired antibiotic resistance genes, including those for clindamycin. The DMEA09 strain did not show hemolysis as a result of analyzing α- and β-hemolysis. It did not form biofilm either. A genomic analysis revealed that strain DMEA09 did not encode for any virulence factors including hemolysin. Most importantly, multilocus sequence typing revealed that the clonal group of strain DMEA09 was distinguished from clinical isolates. Regarding its technological properties, strain DMEA09 could grow in the presence of 6% salt. It showed protease activity when the salt concentration was 3%. It did not exhibit lipase activity. Its genome possessed 37 putative protease genes and salt-tolerance genes for survivability under salt conditions. Consequently, strain DMEA09 shows safe and technological properties as a new starter candidate. This was confirmed by genome analysis.

## 1. Introduction

The genus *Enterococcus* is known to play an important role in the intestinal symbiosis of humans and animals [1]. It is used as an indicator of fecal contamination along with *E. coli* [2]. Recent studies have reported that enterococci are common nosocomial pathogens, causing diseases such as bacteremia, urinary tract infections, endocarditis, surgical site infections, root canal failure, immune evasion, and tissue damage in humans [3,4]. Virulence factors such as enterococcal surface protein, collagen adhesion, pheromone expression, and cytolysin of enterococcal isolates from hospitals have also been reported [5,6,7]. *Enterococcus* species can easily acquire antibiotic resistance genes, plasmids, and virulence factors when exposed to selective antibiotic pressure via horizontal transfer [8,9]. These results show that enterococci have potential as pathogenic bacteria.

Ironically, enterococci have a longer history of being isolated from fermented foods than being known as pathogenic bacteria. They have been detected in various environments (such as soil, plants, and water) and foods (such as fermented sausage, cheese, picked fruit, vegetables, and fermented soybeans) [10]. Enterococci contribute to the sensory development and ripening of fermented foods such as cheese, sausages, and *doenjang* [11,12,13]. Therefore, they have been used as starter cultures for fermenting food products [14,15]. Many enterococcal strains are used industrially, such as for bacteriocin production [15,16,17,18]. Some are commercially available as probiotics to improve animal health, including *Enterococcus faecium* SF68 (Cerbios-Pharma SA, Barbengo, Switzerland) and *Enterococcus faecalis* Symbioflor 1 (SymbioPharm, Herborn, Germany) [19,20]. In addition, enterococci from fermented food show low antibiotic resistance prevalence [21,22,23], while the clinical isolates of enterococci show high frequencies of antibiotic resistance and multidrug resistance [24,25,26,27]. Genomic analysis has revealed that *Enterococcus* species have very few intrinsic virulence traits [28,29,30].

Previously, we found that *Enterococcus faecium* is a dominant species in *meju*, a Korean traditional dried fermented soybean brick. *E. faecium* displays protease and lipase activities [31]. *E. faecium* 0AME20 was selected as a starter candidate through safety assessments (such as antibiotic susceptibility, hemolysis, and biofilm formation) and its functional properties [22]. A safety evaluation experiment found that food-derived *Enterococcus* not only showed a low frequency of antibiotic resistance, but also showed benign characteristics without causing hemolysis [22]. When the multilocus sequencing typing technique was applied to check the genotype according to the separation source, it was confirmed that *Enterococcus* including *E. faecium* derived from food could be distinguished from genotypes with other separation sources such as animals and the environment [22]. Above all, the selected *E. faecium* 0AME20 as a starter candidate contributed to the production of volatile compounds in in vitro and in situ soybean fermentation [11,12]. These results indicate that food-derived *E. faecium* are basically non-pathogenic and that they can contribute to the flavor formation of fermented soybeans as starter bacteria for fermenting soybeans. Continuous efforts are needed to select bacteria for fermented foods. In this study, a novel strain, *Enterococcus faecium* DMEA09, was selected to expand the starter pool. The aim of this study was to determine its safety aspect at the genetic level through genetic analysis. In addition, the existence of a gene associated with horizontal gene transfer was confirmed and the possibility of using it as a starter candidate for fermenting food was highlighted.

## 2. Materials and Methods

### 2.1. Bacterial Strains and Culture Conditions

*Enterococcus faecium* DMEA09 was originally isolated from fermented *meju* [32]. It was selected as a novel starter candidate and subjected to in vitro experiments and genomic analysis. *E. faecium* KCCM 12118^T^ was used as a control to compare its phenotypical properties and the genome. *E. faecium* strains were cultured in tryptic soy agar (TSA; Becton, Dickinson and Co., Sparks, MD, USA) at 37 °C for 18 h to maintain their bacterial traits.

### 2.2. Safety Assessment

A safety assessment was performed based on antibiotic susceptibility and hemolysis. Antibiotic susceptibility was checked by determining antibiotic minimum inhibitory concentrations (MICs) using the broth microdilution method [33]. Each of ten antibiotics (ampicillin, chloramphenicol, clindamycin, erythromycin, gentamicin, kanamycin, streptomycin, tetracycline, tylosin, and vancomycin) was serially diluted (two-fold) with deionized water to obtain working dilutions. The final concentration of the antibiotic in each well of a 96-microwell plate ranged from 0.5 mg/L to 1024 mg/L. Bacterial strain preparation and inoculation were performed in the same manner as described in a previous paper [22]. The MIC was recorded as the lowest concentration where no growth was observed in wells after incubation at 37 °C for 15 h. The MIC results were confirmed by at least three independently performed tests. Strains with MICs higher than the breakpoint were considered resistant [34].

Hemolysis was confirmed by α- and β-hemolytic activity tests. Briefly, 5% (*v*/*v*) rabbit blood (MB Cell, Seoul, Republic of Korea) for α-hemolysis and 5% (*v*/*v*) sheep blood (MB Cell) for β-hemolysis were supplemented into TSA. After inoculating the strain on the medium, hemolysis was determined based on the presence or absence of clear zone formation. The α-hemolytic activity was determined by incubating at 37 °C for 24 h and the β-hemolytic activity was determined by cold shock at 4 °C for 24 h after incubation at 37 °C for 24 h [35]. *Staphylococcus aureus* USA300-p23 and *S. aureus* RN4220 were used as positive and negative controls for hemolytic analyses, respectively [36]. The experiments were conducted at least three times on separate days.

According to the previous method [22], an overnight culture of *S. aureus* RN4220, USA300-p23, and DMEA09 in tryptic soy broth (TSB; Becton, Dickinson and Co.) was inoculated at 1% (*w*/*v*) into fresh TSB. Culture (200 μL) was added to each well of a 96-well microtiter plate and incubated for 24 h at 37 °C without shaking. After the supernatant was discarded, the plates were dried, and the cells were stained with 0.1% crystal violet.

### 2.3. Technological Assessment

The salt tolerance of *E. faecium* strains was determined by examining growth on TSA supplemented with NaCl at a final concentration of 0.5–9% (*w*/*v*). Growth on 0.5%, 3%, 6%, and 9% NaCl was observed after 1, 2, 3, and 4 days of incubation. The protease and lipase activities were determined on TSA containing 2% (*w*/*v*) skim milk and tributyrin-agar (Sigma-Aldrich, Burlington, MA, USA) containing 1% (*v*/*v*) tributyrin, respectively. The effect of NaCl was determined by adding NaCl to the appropriate medium. The enzyme activities were determined by clear halo formation around colonies.

### 2.4. Genome Sequencing

A total of 136,666 reads (5355.39 × coverage) were generated. These reads were assembled with the HGAP4 algorithm in SMRT Link (version 10.1.0; Pacific Bioscience). Genome annotation was performed using the NCBI Prokaryotic Genome Annotation Pipeline (version 4.6) [37]. Open reading frames (ORFs) were predicted using Glimmer 3 [38], followed by annotation through a search against the Clusters of Orthologous Groups (COG) database [38]. 

The genomic DNA of the *E. faecium* strain DMEA09 was isolated and purified using a MagAttract HMW DNA Kit (QIAGEN, Hilden, Germany). The concentration and purity of the extracted DNA were determined using a Qubit 2.0 fluorometer (Invitrogen, Carlsbad, CA, USA). Whole-genome sequencing was performed using a PacBio_10K sequencing system (Pacific Bioscience, Menlo Park, CA, USA) by CJ Bioscience, Inc. (Seoul, South Korea). One contig was obtained from a total of 106,168 reads (2897.12 × coverage) generated. The reads were assembled with the HGAP4 algorithm in SMRT Link (version 10.1.0; Pacific Bioscience). Gene annotation was performed using the NCBI Prokaryotic Genome Annotation Pipeline (version 4.6) [37]. The gene functions were analyzed by performing a search against the Clusters of Orthologous Groups (COG) database [38] and SEED database (https://rast.nmpdr.org/rast.cgi, accessed on 25 August 2022).

### 2.5. Comparative Genome Analysis

For genome comparison, the genome of the type strain KCCM 12118^T^ (GenBank Accession No. GCA_015767695.1) was retrieved from the National Center for Biotechnology Information (NCBI) database (http://ncbi.nlm.nih.gov/genomes, accessed on 25 November 2020). The average nucleotide identity (ANI) was used to check the similarity of the core genome [39]. Core-genome and pan-genome analyses were performed using the Efficient Database framework for comparative Genome Analyses with BLASTP score Ratios (EDGAR) [40]. Rapid Annotation using Subsystem Technology (RAST) [41] and Interactive Pathways Explorer v3 (https://pathways.embl.de, accessed on 30 June 2023) were used to determine the gene contents based on functional subsystem classifications and to estimate the amino acid metabolic pathways. Comparative analyses at the protein level were performed with an all-against-all comparison of the annotated genomes. 

### 2.6. Antibacterial Activities against Pathogenic Bacteria

The antibacterial activities of strain DMEA09 against nine foodborne pathogenic bacteria, *Bacillus cereus* KCCM 11341, *Enterococcus faecalis* KCTC 2011, *Listeria monocytogenes* ATCC 13932, *Staphylococcus aureus* ATCC 12692, *Alcaligenes xylosoxidans* KCCM 40240, *Flavobacterium lutescens* KCCM 11374, *Escherichia coli* O157:H7 EDL 933, *Vibrio parahaemolyticus* ATCC 12802, and *Salmonella enterica* KCCM 11862 were determined using the agar well diffusion method. Pathogens as indicator strains from an overnight culture in TSB were inoculated at 1% (*v*/*v*) into fresh TSB media and incubated to an OD600 of 1.0. Then 200 μL of each culture was poured onto TSA. A hole with a diameter of 6 mm was punched aseptically with a sterile cork borer and 20 μL of concentrated solution of *E. faecium* was introduced into the well. The agar plates were then incubated at 37 °C for 18 h. *E. faecium* concentrate was cultured at 37 °C in TSB broth for 18 h. The supernatant was obtained through centrifugation (1 min). It was then concentrated four times using HyperVAC (HyperVAC-MAX, Hanil Scientific Inc., Gimpo, Republic of Korea). The relative size of the zone of clearing around the punched hole was used as an indicator of antibacterial activity. 

### 2.7. Multilocus Sequence Typing and Phylogenetic Analysis

Multilocus sequence typing (MLST) of the DMEA09 strain was performed according to the previous published *Enterococcus* MLST scheme using seven housekeeping genes: *adk* (adenylate kinase), *atpA* (ATP synthase subunit alpha), *ddl* (D-alanine—D-alanine ligase), *gdh* (glucose-6-phosphate dehydrogenase), *gyd* (glyceraldehyde-3-phosphate dehydrogenase), *pstS* (Phosphate-binding protein PstS), and *purK* (N5-carboxyaminoimidazole ribonucleotide synthase) [42]. The internal regions of these seven genes were obtained and combined manually in the order of *atpA, ddl, gdh, purK, gyd, pstS,* and *adk* using LaserGene 7.1 software (DNASTAR, Madison, WI, USA). A phylogenetic analysis of combined sequences was performed using the maximum likelihood method in MEGA 11 [43,44]. The bootstrapping values were estimated from 1000 repeated calculations. The reference strain gene sequences were obtained from NCBI. 

### 2.8. Statistical Analysis

Duncan’s multiple range test following a one-way analysis of variance (ANOVA) was used to evaluate significant differences between the average values of MIC. Values with *p* < 0.05 were considered statistically significant. All statistical analysis was performed using the SPSS software package (version 27.0; SPSS, IBM, Armonk, NY, USA).

### 2.9. Nucleotide Sequence Accession Number

The complete genome sequence of *E. faecium* DMEA09 (0SSM4) was deposited in the DDBJ/ENA/GenBank under accession number NZ_CP115812.1.

## 3. Results and Discussion

### 3.1. Phenotypic Properties of Enterococcus faecium DMEA09 as a Starter Candidate

A starter candidate should be safe. However, the technological properties that contribute to fermentation and affect the sensorial properties or physical properties of fermented foods are also important. Therefore, safety and functionality for fermentation were evaluated for the isolated strain, DMEA09.

#### 3.1.1. Safety Properties of *Enterococcus faecium* DMEA09

A virulence factor for *E. faecium* has not yet been clearly determined. However, for Qualified Presumption of Safety (QPS) registration with the European Union Food Safety Authority (EFSA), or even if it is registered, its antibiotic resistance and toxic activity should be determined [45]. Therefore, acquired antibiotic resistance and hemolysis were analyzed for *E. faecium* DMEA09. Additionally, its biofilm formation ability was determined.

The type of antibiotic used to evaluate antibiotic susceptibility to microorganisms as starter candidates has not been determined, internationally. However, the EFSA has suggested guidelines for assessing antibiotic susceptibility to microorganisms including *E. faecium* used in food and feed [34], and antibiotic susceptibility was checked according to those guidelines. We evaluated the antibiotic resistance of *E. faecium* DMEA09 to 10 antibiotics (ampicillin, chloramphenicol, clindamycin, erythromycin, gentamicin, kanamycin, streptomycin, tetracycline, tylosin, and vancomycin) based on the presented EFSA guidelines. The DMEA09 strain was found to be sensitive to nine of ten antibiotics (except clindamycin) by showing lower MIC values than the cut-off values suggested by the EFSA for the ten antibiotics (Table 1). The MIC value of clindamycin was 8 mg/L, which was twice as high as the breakpoint of 4 mg/L. 

Since there are no guidelines for identifying virulence factors for *E. faecium* yet, hemolysis and biofilm formation, generally assessed for safety, were evaluated (Figure 1) [35,36]. *E. faecium* DMEA09 did not show hemolysis as a result of analyzing α- and β-hemolysis in the medium including rabbit and sheep blood, contrary to the positive control *S. aureus* USA300-p23 (Figure 1A,B). In addition, *E. faecium* DMEA09 showed no ability to form biofilm (Figure 1C). The above results indicate that *E. faecium* DMEA09 is sensitive to nine of ten antibiotics (except clindamycin), non-hemolytic, and non-biofilm forming. 

#### 3.1.2. Technological Properties of *Enterococcus faecium* DMEA09

Fermentation breaks down macromolecules into micromolecules through enzymatic action. The decomposed ingredients contribute to aroma, taste, and structure [46]. *E. faecium* as a starter candidate increases sensory properties in yogurt or cheese fermentation [47,48]. Also, in our previous experiments, it was confirmed that *E. faecium* showing protease activity increases the production of volatile fragrance components in soybean fermentation [11,12]. These results are necessary to specify the protease gene that contributes to the volatile compounds. In any case, *E. faecium* contributes to the sensory properties, which are affected by enzyme activity, of fermented foods. Fermented foods often have a high salt content. Thus, the presence or absence of resistance to salt is sometimes determined as the fermentation suitability of a species. Therefore, in this study, an experiment was conducted to determine the salt resistance of strain DMEA09 along with its protease, lipase, and amylase activities for decomposing proteins, lipids, and carbohydrates. Strain DMEA09 was grown in the presence of 6% salt. It showed protease activity when the salt concentration was 3% (Figure 2). However, it did not show any lipase activity (Appendix A). In conclusion, strain DMEA09 is phenotypically safe in terms of protease activity. It can grow in the presence of salt up to 6%. Thus, it is suitable as a starter of fermented food with a lot of proteins and a high salt concentration. 

### 3.2. General Genome Properties of Enterococcus faecium Strain DMEA09

To gain insight into the genetic background of starter candidate strain DMEA09, its complete genome sequence was determined. The complete genome of *E. faecium* DMEA09 is a circular 2,546,756-bp chromosome with a G + C content of 38.53 mol%. Sixty-eight tRNA genes and eighteen rRNA genes were identified in the genome. The ANI of the DMEA09 genomic sequence was 95.1% with *E. faecium* KCCM 12118^T^. 

Genomic analysis predicted 2433 open reading frames. Of them, 2264 genes were assigned functionally to categories based on the COG database. Excluding the category of “function unknown”, “carbohydrate transport and metabolism” was the most abundant COG category (303 genes, 12.9%), followed by “replication, recombination and repair” (196 genes, 8.3%) and “transcription” (195 genes, 8.3%) (Figure 3A and Appendix A). The SEED subsystem categorized 935 genes. “Protein Metabolism” (181 genes, 19.4%) was the most abundant subsystem category, followed by carbohydrates (149 genes, 15.9) (Figure 3B and Appendix A).

### 3.3. Genome-Based Safety Insights of Technological Properties of Strain DMEA09

The EFSA has introduced the QPS approach to check the safety of microorganisms throughout the food chain [49]. However, *Enterococcus* is one of the bacteria that causes local or hospital infections. Virulent strains in *Enterococcus* have been reported, including *E. faecium.* This genus has not been listed on the QPS list [45]. The obvious reason for not being on the QPS list was the lack of clear information about the virulence factor of *Enterococcus* [48]. The EFSA suggests that enterococci are not safe at the genus level. Because it is difficult to discuss safety at the species and genus levels, it is appropriate to review safety by strain. Based on QPS, *E. faecium* DMEA09 has a clear taxonomy at the species level. The usage history of *E. faecium* is also sufficient with the frequency and amount detected in existing probiotics or fermented foods [19,20]. Since toxic factors such as enterococcal surface protein, collagen adherence, pheromone expression, and cytolysin were mentioned as reasons for not registering enterococci in QPS [50], the genome of strain DMEA09 was analyzed. As a result, those virulence factors were not found to be possessed. In addition, toxin and hemolysin genes were not identified from the annotated genes of the DMEA09 genome by searching for keywords.

In addition to these factors, QPS requires information about antibiotic resistance ability. This can be replaced by antibiotic sensitivity to the ten antibiotics suggested by EFSA in the experimental results. The presence of genes was also confirmed. In addition to these genes, QPS also requires the absence of acquired antibiotic resistance ability. Although antibiotic resistance does not directly contribute to virulence, it can indirectly increase virulence via quorum sensing [51]. Furthermore, antibiotic resistance genes can be transferred to other bacteria during fermentation. They can also be transferred to intestinal commensal bacteria as they pass through the intestine [52,53]. Therefore, it is essential to check for the presence of acquired antibiotic resistance genes. According to the antibiotic susceptibility of EFSA guidelines [34], *E. faecium* DMEA09 was sensitive to 9 out of 10 antibiotics (except clindamycin) tested. In the case of clindamycin, the MIC value (8 mg/L) was twice as high as the break point (4 mg/L) (Table 1). One of the factors that confers resistance to clindamycin occurs through the methylation of ribosomal target sites bound by clindamycin, and the representative gene is the erythromycin ribosome methylase (*erm*) gene [54]. However, this gene was not identified in DMEA09 genome. We confirmed the presence of the spermidine acetyltransferase (*speG*) gene in DMEA09 genome because we reported in previous experiments that it showed weak resistance to clindamycin [55], which was also not detected. In the case of affecting antibiotic resistance, specific antibiotic resistance genes may occur through changes in antibiotics or antibiotic binding sites as above, but they also show weak resistance due to environmental factors such as salt concentration [56]. In the case of clindamycin-specific resistance, the MIC value clearly increases, but it is weakly resistant due to environmental factors. The MIC to clindamycin of DMEA09 increased weakly and no resistance gene to clindamycin was detected; thus, it is suggested that it has non-specific resistance. Consequently, these results confirm that the strain DMEA09 is a safe strain not only based on phenotype analysis, but also based on genomic analysis results. 

Unlike clinical isolates, *E. faecium* derived from food including *E. faecium* DMEA09 non-detecting toxic factor has a low frequency of detection of toxic actors and a low rate of resistance to antibiotics [21,22,23,25,26,27]. Therefore, it is necessary to determine whether *E. faecium* bacteria are pathogens or starter candidates considering safety concerns. In addition, whether the toxic factors mentioned above are directly related to pathogenicity should be determined. Collagen is a structural protein that is widely distributed in animal cells. Bacteria with collagen adherence abilities are known to have an effect at the beginning stage or persistence stage of bacterial infection [57]. At the same time, collagen adhesion produced by bacteria that are considered healthy, such as lactic acid bacteria, shows a probiotic effect by preventing pathogenic bacteria from settling down [58,59]. As such, the pathogenicity of collagen adherence is different depending on the species in question. Like the collagen adhesion gene, enterococcal surface protein is also interpreted differently depending on the species. The surface protein of pathogenic bacteria is known to increase virulence by contributing to biofilm formation [60]. The surface protein of lactic acid bacteria has been reported to contribute to reduced pathogen intestinal colonization [61]. In conclusion, there are in vitro results for the potential virulence of enterococcal surface protein (*esp*), collagen adhesion (*ace*), and pheromone expression (*ccf*) known as the virulence factor of *Enterococcus faecium*. However, there are no direct pathogenicity results such as in vivo experiments. Therefore, research on whether the gene is directly involved in pathogenicity is necessary.

### 3.4. Sequence Type of Strain DMEA09 Using Multilocus Sequence Typing

As a result of phenotype analysis, DMEA09 was found to be an antibiotic-sensitive strain without causing hemolysis or forming biofilm. The results of genomic analysis supported the results of phenotype analysis. Nevertheless, concern about the safety of *E. faecium* continued to be raised. Accordingly, whether *E. faecium* derived from food could be distinguished from clinical isolate was determined. Whether genotypes could be distinguished was also determined by applying existing pubMLST.

MLST is a highly discriminative typing method. It is appropriate for genetically coherent organisms to provide a measure of genetic relatedness among strains [62]. MLST has a significant advantage of worldwide comparison using exchangeable data based on DNA sequence results. In addition, MLST has been used to study the long-term epidemiology of bacterial species. It has provided insights into population structure and patterns of evolutionary descent [63]. Our goal was to investigate genetic differences among *E. faecalis* isolates from fermented food, animals, and the environment using MLST for starter development. MLST is a suitable method for distinguishing between safe *E. faecium* derived from food and *E. faecium* isolated from hospitals.

As of April 2023, 3188 genomes of *E. faecium* were registered in the NCBI database. There are 307 cases of complete genome sequence. Among them, 10 genomes of food-derived strains, including DMEA09 strains, were obtained (Figure 4). For comparison, 10 strains derived from humans and clinical isolates were randomly selected (Figure 4). Using a database of seven housekeeping genes registered in pubMLST, the allele number of each strain was identified and the sequence type was identified by a combination of these numbers. As a result, the DMEA09 strain corresponded to sequence type (ST) 695 (Figure 4). When applying the eBURST algorithm based on ST, it was divided into two clonal complexes and five singletons. Surprisingly, clinical isolates were included in the CC17 group. CC17 is known as a clinical complex consisting of STs related to hospital-associated infection, acquiring antimicrobial resistance [64]. Furthermore, five strains determined to have food origin corresponded to CC94. These strains corresponding to CC94 were strains derived from Korea and China. It was confirmed that they were distinguishable from clinical isolates. These results show that the strain corresponding to CC94 is distinguished from CC17, which mainly consists of clinical isolates, and we assumed that strain corresponding to CC94 should be safe for toxicity factors and antibiotic resistance, unlike clinical isolation strains.

In the case of *E. faecium*, it is difficult to distinguish whether it is dangerous or safe based on species. For example, although *E. coli* is known to cause food poisoning, it is mostly harmless. It is one of the healthy intestinal microorganisms, although some are pathogens, causing diarrhea, urinary tract infections, and respiratory diseases. *Enterococcus faecium* is also mostly harmless. It is known as an intestinal bacterium. The results of this experiment confirm that *E. faecium*, especially food-derived *E. faecium* of the DMEA09 strain, is safe through both phenotype and genome analyses. At the same time, the MLST results suggest that the sequence type of strain DMEA09 can be used as one of the methods for identifying safe *E. faecium* by confirming that it belongs to MLST genotype CC94.

### 3.5. Genome-Based Technological Properties of Strain DMEA09

The DMEA09 strain showed growth and protease activity in the presence of 6% and 3% (*w*/*v*) NaCl, respectively (Figure 2). The DMEA09 genome possesses 30 protease genes assigned an E.C. number and 7 protease genes not assigned an E.C. number (Table 2). To achieve salt tolerance, DMEA09 possesses the osmoprotectant uptake (Opu) systems OpuB, and OpuD as well as biosynthetic genes for the compatible solutes, glycerol, glutamate, cardiolipin, and proline (Figure 5). Thus, we assumed that genes associated with protease and salt tolerance translated into stronger protease activity under NaCl pressure (Figure 2).

A genomic analysis of *E. faecium* DMEA09 revealed the safety of DMEA09 as well as phenotypic results. Moreover, the strain DMEA09 showed enzymatic and antibacterial activities as well as salt tolerance, which could contribute to the sensory properties of fermented food as a potential starter candidate. Genome analysis can help us explain phenotypic traits. Finally, the CC of *E. faecium* DMEA09 based on MLST was distinguished from clinical isolates. These results provide the genetic basis for further comparative, functional, and safety genomic analyses to help select potential starter candidates for use in food, animal feed, and medical industries.

Consequently, the safety and technological properties of *Enterococcus faecium* DMEA09 derived from *meju* as a starter candidate for food fermentation were determined. In addition, genomic analysis was performed to determine its safety and technological activities. Strain DMEA09 showed sensitivity to nine antibiotics (ampicillin, chloramphenicol, erythromycin, gentamicin, kanamycin, streptomycin, tetracycline, tylosin, and vancomycin) and did not show hemolysis and biofilm. However, the MIC of DMEA09 against clindamycin showed two-fold the breakpoint of the EFSA. However, genome analysis showed the non-possession of the clindamycin resistance gene as well as the acquired antibiotic resistance and toxin genes. These results suggest a safe strain, and the proteolytic activity of the DMEA09 strain is expected to contribute to the production of amino acids by decomposing proteins in fermented foods. It has been known that the enzyme activity of a starter strain improves the sensory properties of fermented foods [65,66,67]. However, studies that specify enzymes that contribute to sensory properties are difficult to find. Strain DMEA09 showed protease activity at a 3% salt concentration. Also, strain 0AME20, isolated in a previous study, showed lipase activity at 2% salt concentration as well as protease activity at 3% salt concentration [22]. These results assume that when the two strains are applied to fermented foods as starter strains, the effect of lipase can be analyzed. As a result, it is significant that DMEA09 strain has been secured as a safe starter candidate for fermented foods.

## Figures and Tables

**Figure 1 foods-12-03008-f001:**
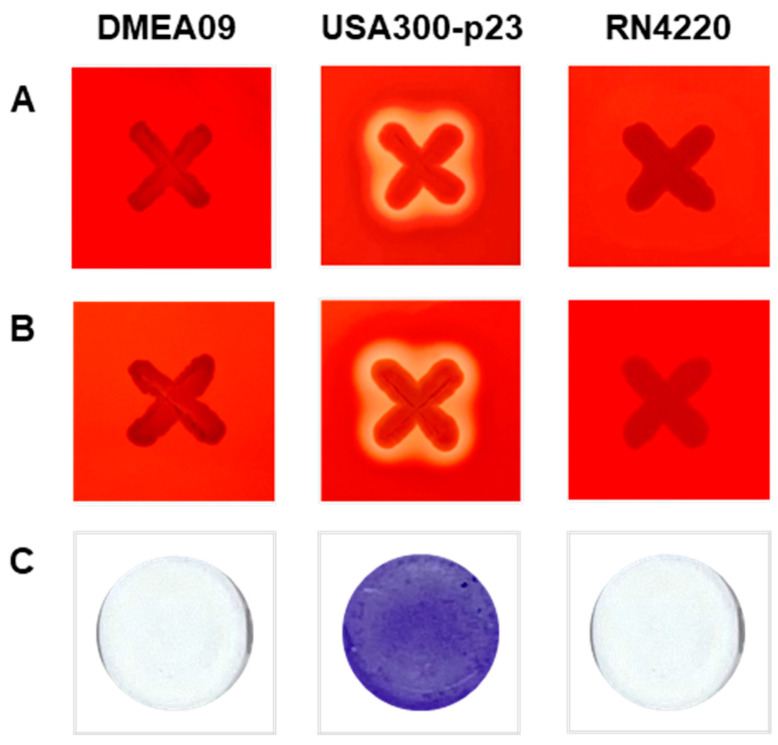
α-Hemolytic activity (**A**), β-hemolytic activity (**B**), and biofilm formation (**C**) of *Enterococcus faecium* DMEA09. *Staphylococcus aureus* strains USA300-p23 and RN4220 were used as positive and negative controls, respectively.

**Figure 2 foods-12-03008-f002:**
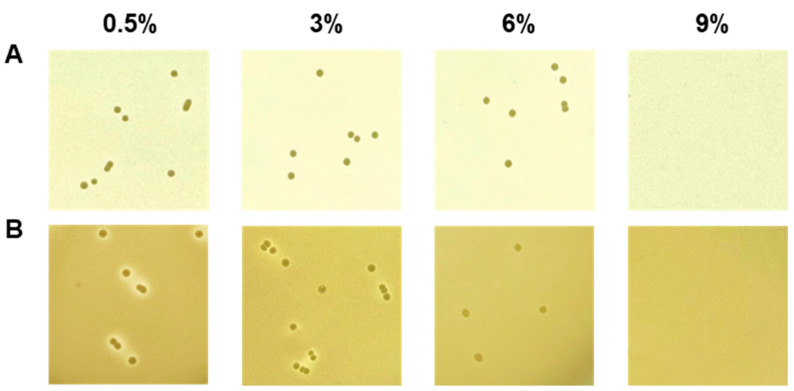
Growth (**A**) and proteolytic activity (**B**) of *Enterococcus faecium* DMEA09 on media supplemented with 0.5%, 3%, 6%, and 9% NaCl.

**Figure 3 foods-12-03008-f003:**
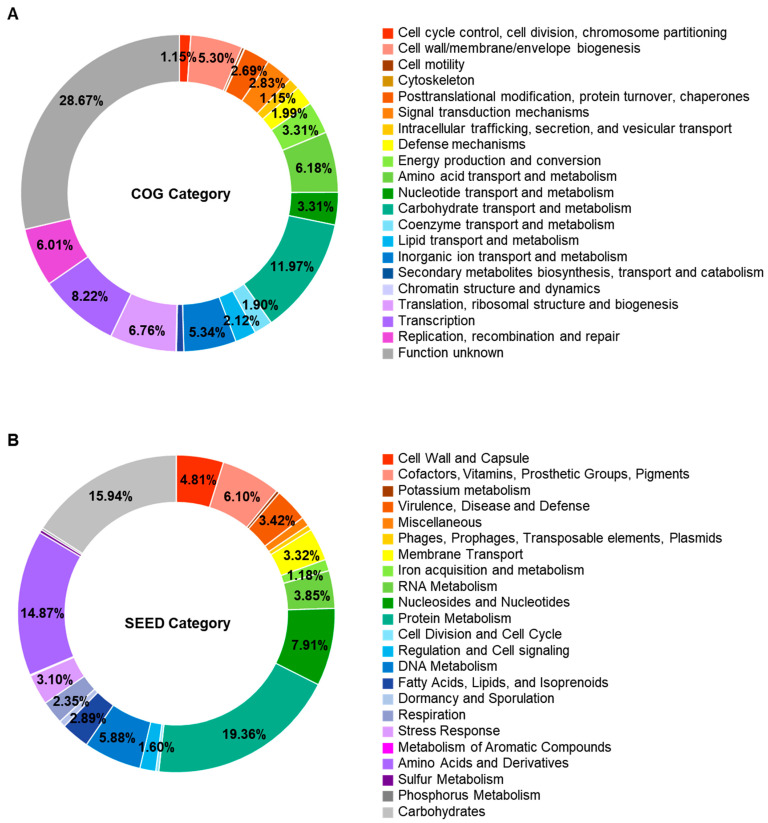
Functional classification through Clusters of Orthologous Groups database (**A**) and SEED subsystem (**B**) of *Enterococcus faecium* DMEA09. Only those with a relative ratio of 1% or more are displayed in the circle.

**Figure 4 foods-12-03008-f004:**
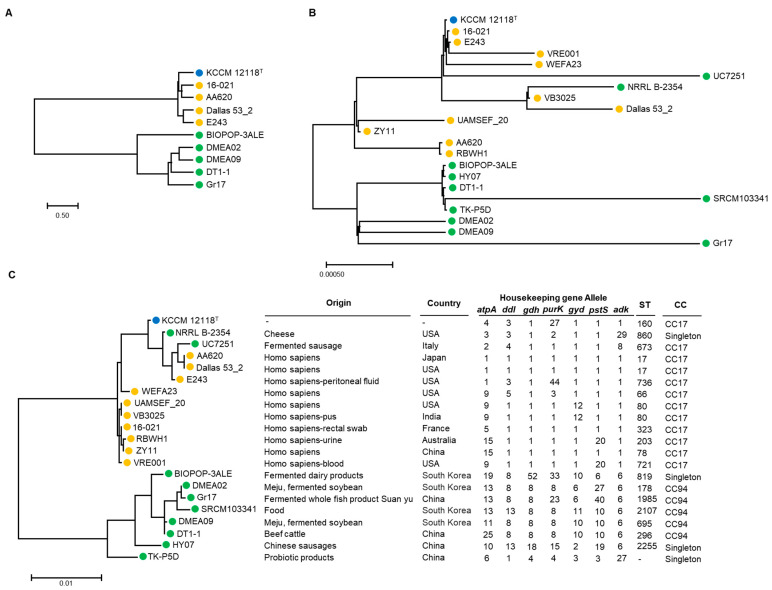
Phylogenetic tree of ANI (**A**), 16S rDNA (**B**), and MLST and housekeeping gene allele (**C**) using the maximum likelihood method in MEGA 11 and OrthoANI Tool ver. 0.93.1. Blue, orange, and green circles indicate that isolates are from type, clinical, and food strains, respectively.

**Figure 5 foods-12-03008-f005:**
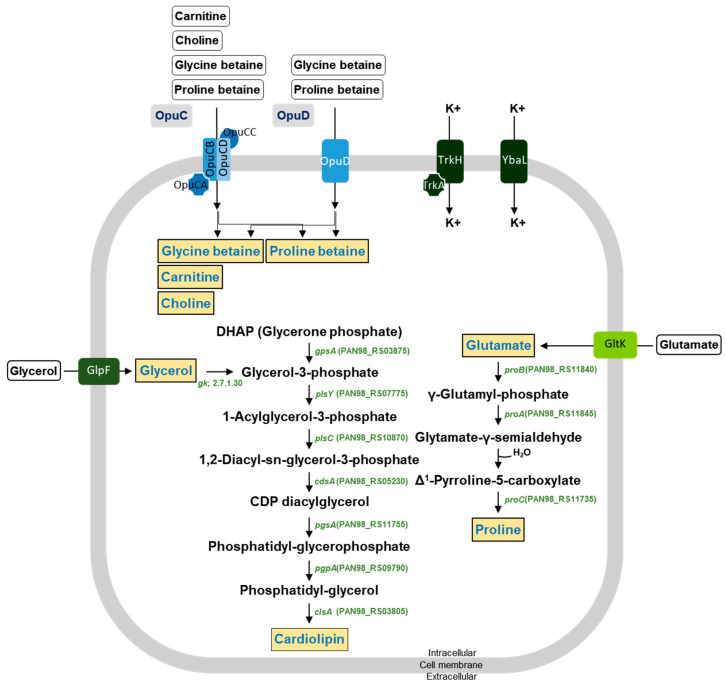
Predicted membrane transport systems and synthesis pathways for osmoprotectants in the *E. faecium* DMEA09 genome. Osmoprotectants are depicted in orange. Black arrows correspond to potential enzymatic reactions catalyzed by gene products encoded by the three genomes.

**Table 1 foods-12-03008-t001:** Minimal inhibitory concentrations (mg/L) against ten antibiotics of *E. faecium* DMEA09.

	DMEA09	Breakpoint *
Ampicillin	2	2
Chloramphenicol	8	16
Clindamycin	8	4
Erythromycin	4	4
Gentamicin	4	32
Kanamycin	8	1024
Streptomycin	64	128
Tetracycline	1	4
Tylosin	1	4
Vancomycin	1	4

*: EFSA breakpoint for *Enterococcus faecium* [34]. Statistical relevance was analyzed using Duncan’s multiple range test.

**Table 2 foods-12-03008-t002:** Annotated protease genes identified in the DMEA09 genome.

Gene Locus	Product	E.C. No.	COG
PAN98_RS01200	Glutamyl aminopeptidase	3.4.11.7	S
PAN98_RS01220	Trypsin-like peptidase-domain-containing protein	3.4.21.107	M
PAN98_RS02400	SOS-response-associated peptidase family protein		OU
PAN98_RS02525	Peptidase M13	3.4.24.-	O
PAN98_RS02955	M20 family metallopeptidase		S
PAN98_RS03050	Pyroglutamyl-peptidase I	3.4.19.3	E
PAN98_RS03895	Xaa-Pro peptidase family protein	3.4.13.9	MU
PAN98_RS04325	C69 family dipeptidase	3.4.-.-	E
PAN98_RS05090	Rhomboid family intramembrane serine protease	3.4.21.105	NU
PAN98_RS05235	RIP metalloprotease RseP	3.4.24.-	M
PAN98_RS05520	Signal peptide peptidase SppA	3.4.21.-	O
PAN98_RS05540	ATP-dependent Clp protease ATP-binding subunit ClpX		O
PAN98_RS05875	Peptidase T	3.4.11.4	U
PAN98_RS06780	Signal peptidase II	3.4.23.36	O
PAN98_RS06880	Carboxypeptidase M32	3.4.17.19	J
PAN98_RS07560	M15 family metallopeptidase	3.4.17.14	E
PAN98_RS07790	ATP-dependent protease ATPase subunit HslU		E
PAN98_RS07795	HslU--HslV peptidase proteolytic subunit	3.4.25.2	U
PAN98_RS07840	Signal peptidase I	3.4.21.89	E
PAN98_RS08710	ATP-dependent Clp protease ATP-binding subunit		O
PAN98_RS08980	Type I methionyl aminopeptidase	3.4.11.18	E
PAN98_RS08995	Aminopeptidase	3.4.11.-	U
PAN98_RS09225	Xaa-Pro peptidase family protein	3.4.13.9	S
PAN98_RS09305	Signal peptidase I	3.4.21.89	S
PAN98_RS09330	M42 family metallopeptidase	3.4.11.-	V
PAN98_RS09600	ATP-dependent Clp endopeptidase proteolytic subunit ClpP	3.4.21.92	S
PAN98_RS09710	C1 family peptidase	3.4.22.40	E
PAN98_RS09850	Zinc metallopeptidase	3.4.21.89	NU|M
PAN98_RS09900	Beta-aspartyl-peptidase	3.4.19.-	O
PAN98_RS10490	Type II CAAX endopeptidase family protein		E
PAN98_RS10830	Oligoendopeptidase F	3.4.24.-	M|S
PAN98_RS11525	ATP-dependent zinc metalloprotease FtsH	3.4.24.-	O
PAN98_RS11590	M3 family oligoendopeptidase	3.4.24.-	O
PAN98_RS12075	A24 family peptidase	3.4.23.43, 2.1.1.-	NOU
PAN98_RS12200	U32 family peptidase	3.4.-.-	O
PAN98_RS12205	Peptidase U32 family protein	3.4.-.-	O
PAN98_RS12210	ATP-dependent Clp protease ATP-binding subunit		O

## Data Availability

The data used to support the findings of this study can be made available by the corresponding author upon reasonable request.

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
