# Peer review of "Novel Starter Strain Enterococcus faecium DMEA09 from Traditional Korean Fermented Meju"

_foods, 2023, doi:10.3390/foods12163008_

Round 1

Reviewer 1 Report

Comments and Suggestions for Authors

In this study, the authors did a lot of research on Enterococcus faecium DMEA09, including phenotype analysis and genomic analysis. It is a valuable study for reference. The discussion of results is insufficient and the representativeness of assessment methods is not fully discussed. These two parts should be supplemented and revised.

#1. The discussion of results is insufficient. Please supplement this part and cite proper references.

#2. Please explain more about why you chose this strain and what the advantages of this strain are.

#3. Why were these 10 antibiotics chosen for the Antibiotic susceptibility method? Can you provide relative reference to support your explanation?

#4. In Line 198-199, it showed “The MIC value of clindamycin was 8 mg/L, which was twice as high as the break point of 4 mg/L”. The result of the analysis of the clindamycin is different from the others, whether there is a toxicity concern, please add an explanation and provide references to support this.

#5. In Line 200-201, it showed “hemolysis and biofilm formation generally assessed for safety were evaluated”. Please cite the proper reference to support it.

#6. In Line 224-226, it showed “It showed pro tease activity when salt concentration was 3% (Figure 2). However, it did not show any lipase activity”. Having the ability to proteolyze may also produce small molecule proteins with poor flavor. The breakdown of fats may also produce small molecule with good flavor. Please add an explanation for only protease activity was evaluated in this manuscript. And explain why the disadvantages of protease are not concerned.

#7. In Line 263-264, it showed “Since toxic factors such as enterococcal surface protein, collagen adherence, pheromone expression, and cytolysin were mentioned as reasons for no registering Enterococci in QPS”, please cite the proper references.

#8. In Line 266-267, it showed “toxin and hemolysin gene were not identified from annotated genes of DMEA09 genome by searching for keywords.”. And in Line 283-284, it showed “low frequency of detection of toxic factors”. Please explain clearly whether there are any toxic factors in the analysis of genes in this manuscript.

#9 In Line 330, it showed “food origin corresponded to CC94”. And in Line 332, it showed “the strain corresponding to CC94 is not safe”. Please explain whether this strain of Enterococcus faecal can be regarded as a safe food and cite the proper reference.

#10 The title of the Figure should be at the bottom.

Comments on the Quality of English Language

Moderate editing of English language required

Author Response

Responses to Reviewer #1

In this study, the authors did a lot of research on Enterococcus faecium DMEA09, including phenotype analysis and genomic analysis. It is a valuable study for reference. The discussion of results is insufficient and the representativeness of assessment methods is not fully discussed. These two parts should be supplemented and revised.

#1. The discussion of results is insufficient. Please supplement this part and cite proper references.

> Thanks for your comments. According to the comments of the reviewer, the discussion on the results was supplemented and references were added (L 203-207, 233-240, 302-318, 367-370, 414-431, 553-556, 568-574, 592-597).

#2. Please explain more about why you chose this strain and what the advantages of this strain are.

 > Thanks for your comments. Although it was presented throughout the results as an advantage as a starter strain for DMEA09, additional advantages were presented for them in the future (L414-431).

#3. Why were these 10 antibiotics chosen for the Antibiotic susceptibility method? Can you provide relative reference to support your explanation?

>The types of antibiotics used in antibiotic resistance experiments have not been accurately presented internationally. However, EFSA presented guidelines for antibiotic resistance to microorganisms used in food or feed, and since the DMEA09 strain used in the experiment is a strain for application to food, the experiment was conducted according to the guidelines. In addition, the content was presented in the existing manuscript, and this time it was revised and added it (L203-207).

#4. In Line 198-199, it showed “The MIC value of clindamycin was 8 mg/L, which was twice as high as the break point of 4 mg/L”. The result of the analysis of the clindamycin is different from the others, whether there is a toxicity concern, please add an explanation and provide references to support this.

> Thanks for your comments. The results were explained by adding the genomic analysis results for the experimental content. In addition, an explanation of the content was added according to the reviewer's suggestion (L302-314).

#5. In Line 200-201, it showed “hemolysis and biofilm formation generally assessed for safety were evaluated”. Please cite the proper reference to support it.

> The results were explained by adding the genetic analysis results for the experimental content. And according to the reviewer's comments, references of the content was added (L214-215).

#6. In Line 224-226, it showed “It showed pro tease activity when salt concentration was 3% (Figure 2). However, it did not show any lipase activity”. Having the ability to proteolyze may also produce small molecule proteins with poor flavor. The breakdown of fats may also produce small molecule with good flavor. Please add an explanation for only protease activity was evaluated in this manuscript. And explain why the disadvantages of protease are not concerned.

 > Thanks for your comments. As the reviewer's comments, we cannot rule out the possibility that unwanted sensory ingredients are produced through protease activity. However, when sensory properties are produced in an undesired direction in fermentation, it is known to be due to the formation of bacteria that are not suitable for the beginning of fermentation or late-fermentation. The E. faecium selected in this experiment is reported to have positive results in fermentation as a dominant bacterium in the fermentation stage. Our previous experiments also showed an increase in sensuality when E. faecium showing protease activity was used as a starter. Therefore, the application of E. faecium is expected to have a positive effect on the sensory characteristics. And I added this information to the manuscript (L233-240). However, the relationship between protease genes and sensory properties will be confirmed in other experiments in the future.

#7. In Line 263-264, it showed “Since toxic factors such as enterococcal surface protein, collagen adherence, pheromone expression, and cytolysin were mentioned as reasons for no registering Enterococci in QPS”, please cite the proper references.

 >Added the reference (L287).

#8. In Line 266-267, it showed “toxin and hemolysin gene were not identified from annotated genes of DMEA09 genome by searching for keywords.”. And in Line 283-284, it showed “low frequency of detection of toxic factors”. Please explain clearly whether there are any toxic factors in the analysis of genes in this manuscript.

 > Revised the sentence (L317-318).

#9 In Line 330, it showed “food origin corresponded to CC94”. And in Line 332, it showed “the strain corresponding to CC94 is not safe”. Please explain whether this strain of Enterococcus faecal can be regarded as a safe food and cite the proper reference.

 > Revised the sentence (L367-370).

#10 The title of the Figure should be at the bottom.

 > Revised it.

Reviewer 2 Report

Comments and Suggestions for Authors

The manuscript entitles "Novel Starter Strain Enterococcus faecium DMEA09 from Traditional Korean Fermented Meju" explains the basic technological and genetic analysis of the previously isolated strain of E. faecium. The technological and functional properties should be extensively incorporated to justify the starter potential. Please refer to the comments below:

Introduction: The background information is required to be updated, please consult the recent studies on enterococcus spp. as a potential candidate for probiotics isolated from various traditional products. Moreover, at the end of abstract and introduction mention the novelty aspect of the study.

Materials and method:

                                   Why authors are specifying only salt tolerance, why not other technological benefits of enterococcus spp. please refer to recently published literature such as:

https://doi.org/10.1590/1678-4324-2022210091

More comprehensive safety assessment should be undertaken.

The authors confined the focus only on genomic rather than the ability of starter culture to act as potential candidate. A comprehensive assessment of starter culture candidate will ensure its application to other food products as well.

A section regarding the statistical analysis should be added in the manuscript.

How about aggregation and colonization ability of the starter?

Although the study was carried out in depth on genetic analysis, authors should correlate the functional capability (technological and functional properties) of the isolate with the detailed genetic analysis.

More relevant discussion needs to be incorporated.

Atleast authors should summarize their findings at the end of the results as a conclusion statement. 

Author Response

The manuscript entitles "Novel Starter Strain Enterococcus faecium DMEA09 from Traditional Korean Fermented Meju" explains the basic technological and genetic analysis of the previously isolated strain of E. faecium. The technological and functional properties should be extensively incorporated to justify the starter potential. Please refer to the comments below:

 > Thanks for your good evaluation.

Introduction: The background information is required to be updated, please consult the recent studies on enterococcus spp. as a potential candidate for probiotics isolated from various traditional products. Moreover, at the end of abstract and introduction mention the novelty aspect of the study.

 > Thank you for your opinion. The manuscript is not E. faecium as a probiotic, but a paper as a starter candidate for fermented food. So I mentioned the contents of fermented food in the introduction. Nevertheless, if you give me an opinion that I need to add the content, I will add it later. However, we will review the functionality as a probiotic using the strain in the future, so if it is okay, we will update it in future papers.

Materials and method:

Why authors are specifying only salt tolerance, why not other technological benefits of enterococcus spp. please refer to recently published literature such as:

https://doi.org/10.1590/1678-4324-2022210091

 > Thanks for your comments. In general, LAB is referred to more as a probiotic strain. Probiotics are based on functions that have a positive effect on the human body or animals. We are not referring to strains for this functionality, but rather to selection of starter candidates that play a positive role in food fermentation. Therefore, salt resistance was confirmed because salt is one of the factors that can survive more in fermented foods rather than evaluating factors to survive better in the human body.

More comprehensive safety assessment should be undertaken.

> Thanks for your comments. According to the comments of the reviewer, the discussion on the results was supplemented and references were added (L 203-207, 233-240, 302-318, 367-370, 414-431).

The authors confined the focus only on genomic rather than the ability of starter culture to act as potential candidate. A comprehensive assessment of starter culture candidate will ensure its application to other food products as well.

 > Starter candidates starts with selection. This experiment is about DMEA09 strain selected from many E. faecium isolated from fermented soybeans, meju. Therefore, we have dealt with the safety and technological aspects of the genome-based for the strain. Afterwards, we plan to apply the strain as a starter candidate of fermented food and check their effectiveness.

A section regarding the statistical analysis should be added in the manuscript.

 > We thank the reviewer for pointing this out and we agree with the reviewer. Therefore, the statistical analysis was added (L182-186).

How about aggregation and colonization ability of the starter?

 > We did not conduct experiments on aggregation and colonization because it was not a strain as a probiotic, but an experiment as a starter candidate for fermented foods. In the future, when the strain is applied as a starter candidate of fermented food, the viability of fermented food will be confirmed.

Although the study was carried out in depth on genetic analysis, authors should correlate the functional capability (technological and functional properties) of the isolate with the detailed genetic analysis.

 > We agree with the reviewer. So we presented a possible gene through genetic analysis. However, in such an experiment, the control group must be well selected. For example, a comparison between strains that have enzyme activity and those that do not. This experiment presents basic results, and we believe that we can derive more in-depth results when a control group is selected in the future.

More relevant discussion needs to be incorporated.

> Thanks for your comments. According to the comments of the reviewer, the discussion on the results was supplemented and references were added (L 203-207, 233-240, 302-318, 367-370, 414-431, 553-556, 568-574, 592-597).

Atleast authors should summarize their findings at the end of the results as a conclusion statement. 

 > According to the comments of the reviewer, we added the summary and conclusion (L 414-431).